# Role of Surgical Intervention in Unresectable Non-Small Cell Lung Cancer

**DOI:** 10.3390/jcm9123881

**Published:** 2020-11-29

**Authors:** Shigeki Suzuki, Taichiro Goto

**Affiliations:** 1General Thoracic Surgery, Sagamihara Kyodo Hospital, Kanagawa 252-5188, Japan; shigeki.suzuki@sagamiharahp.com; 2Lung Cancer and Respiratory Disease Center, Yamanashi Central Hospital, Kofu 400-8506, Japan

**Keywords:** non-small cell lung cancer, immunotherapy, unresectable, salvage surgery, oligometastasis, targeted therapy

## Abstract

With the development of systemic treatments with high response rates, including tyrosine kinase inhibitors and immune checkpoint inhibitors, some patients with unresectable lung cancer now have a chance to undergo radical resection after primary treatment. Although there is no general consensus regarding the definition of “unresectable” in lung cancer, the term “resectable” refers to technically resectable and indicates that resection can provide a favorable prognosis to some extent. Unresectable lung cancer is typically represented by stage III and IV disease. Stage III lung cancer is a heterogeneous disease, and in some patients with technically resectable non-small cell lung cancer (NSCLC), multimodality treatments, including induction chemoradiotherapy followed by surgery, are the treatments of choice. The representative surgical intervention for unresectable stage III/IV NSCLC is salvage surgery, which refers to surgical treatment for local residual/recurrent lesions after definitive non-surgical treatment. Surgical intervention is also used for an oligometastatic stage IV NSCLC. In this review, we highlight the role of surgical intervention in patients with unresectable NSCLC, for whom an initial complete resection is technically difficult. We further describe the history of and new findings on salvage surgery for unresectable NSCLC and surgery for oligometastatic NSCLC.

## 1. Introduction

Treatment of patients with lung cancer depends on the histology, tumor stage, molecular characteristics, and assessment of a patient’s overall medical condition. Currently, various guidelines for lung cancer treatment, including those from the American Society of Clinical Oncology, European Society of Medical Oncology, National Comprehensive Cancer Network (NCCN), and The Japan Lung Cancer Society, have been used [1,2]. Patients with stage I-II non-small cell lung cancer (NSCLC) are generally treated with curative-intent surgery if they are operable. Patients with stage III NSCLC are generally treated with a multimodality approach, including surgery, chemotherapy, and radiation therapy (RT). Those with stage IV or recurrent NSCLC are treated with systemic drug therapies, including chemotherapy, tyrosine kinase inhibitors (TKIs), and immune checkpoint inhibitors (ICIs). Molecular-targeted therapies such as TKI are selected if the epidermal growth factor receptor (*EGFR)* gene is mutated, while an ICI and/or cytotoxic chemotherapy is selected if this gene is not mutated.

In general, surgical treatment is selected for tumors that can be completely resected, whereas RT or drug therapy is offered for patients whose tumors cannot be completely resected or who cannot tolerate surgery. In practice, the term “resectable” not only applies to technically resectable, which is “resectable” in a narrow sense, but also refers to cases when resection can be expected to have a favorable prognosis to some extent, which is “resectable” in a broad sense. Although “unresectable” is defined as “unable to be removed using surgery” in the National Cancer Institute dictionary, there is no general consensus regarding the definition of “unresectable” in lung cancer.

Unresectable lung cancer is considered to be represented by stage III and IV disease. Unresectable factors in stage III lung cancer are direct invasion to unresectable organs (T4) or mediastinal/extrathoracic lymph node metastasis (N2/N3). An unresectable feature of stage IV lung cancer is distant metastasis (M1). Regarding mediastinal lymph node metastasis, the 2013 American College of Chest Physicians (ACCP) guidelines defined N2 nodes that have extranodal progression and an invasive nature as infiltrative nodes, while other nodes were defined as discrete nodes. Discrete nodes were considered to likely benefit from surgical therapy [3]. In general, bulky/multi-station/infiltrative nodes are regarded as unresectable, while T4 tumors are tumors sized >7 cm or those invading the mediastinum/heart/diaphragm/carina/trachea/great vessels/recurrent nerve/esophagus/spine, or separate tumor nodule(s) in a different ipsilateral lobe. In patients with unresectable stage III lung cancer, the current standard treatment is concurrent chemoradiotherapy (CRT) [4,5], which provides a median overall survival (OS) of 22–25 months and a 5-year OS of 20% [6]. In this review, we describe the role of surgical intervention in patients with NSCLCs for whom complete resection is technically difficult (“unresectable” in the narrow sense).

## 2. Role of Surgical Intervention in Unresectable Lung Cancer

Stage III lung cancer is a heterogeneous disease, and in some patients with technically resectable NSCLC, including some with cT4/cN2 NSCLC, multimodality treatments, including induction CRT followed by surgery, are a treatment of choice [7]. In practice, induction therapy is used for patients with lung cancer in whom radical resection is difficult at the time of diagnosis but may be expected later owing to therapeutic intervention. The representative surgical intervention for unresectable lung cancer is salvage surgery [8,9]. Although the term “salvage (or “rescue”) surgery” is not clearly defined, it refers to surgical treatment for local residual/recurrent lesions after definitive non-surgical treatment. As a type of surgical intervention for unresectable stage III lung cancer, we describe salvage surgery after definitive CRT or RT.

Although the standard treatment for stage IV lung cancer is drug therapy, there is another choice of treatment for patients with an oligometastatic state. In this state, local therapy for metastatic lesions results in favorable prognosis that is comparable with that in non-metastatic disease. The concept that patients with only a limited number of metastases from a malignant tumor can potentially be cured was developed in 1995 and was termed “oligometastasis,” which describes an intermediate stage between localized and metastasized cancer [10,11]. There is currently no consensus regarding the definition of oligometastatic disease; however, most clinical trial protocols and clinicians accept a definition of 1–3 or 1–5 metastatic lesions [12,13]. Furthermore, in rare cases when definitive systemic therapy is successful, an opportunity for radical resection as salvage surgery is achieved even for stage IV lung cancer [14]. Thus, there are two situations for surgical intervention for stage IV NSCLC: surgery for oligometastatic cases and salvage surgery after definitive systemic therapy (especially, TKI or ICI). The role of surgical intervention for unresectable lung cancer is summarized in Figure 1.

## 3. Salvage Surgery for Stage III NSCLC 

### 3.1. Salvage Surgery after Definitive CRT

The incidence of local recurrence after definitive CRT in patients with stage III NSCLC was 24–35% [15], and the survival rates after CRT were as low as 5–25% [16]. In 2019, Grass et al. showed a high relapse rate after CRT (64%) [17]. Salvage surgery for residual/recurrent tumors is almost the only treatment that can provide a cure. Compared with upfront surgery, salvage surgery after definitive CRT has greater surgical difficulty and a greater possibility of perioperative complications because a high dose of RT strongly affects the target tissue, resulting in more tissue changes [8,9].

To date, there have been limited reports of salvage surgery after definitive CRT for primary lung cancer. Dickhoff et al. reported a systematic review of the literature concerning salvage surgery after definitive CRT for locally advanced NSCLC in 2018 [18]. They reviewed eight papers including 158 patients. For patients undergoing resection (*n* = 152), a total of 44 pneumonectomies, 11 bilobectomies, 89 lobectomies, 6 segmentectomies, and 3 wedge resections were performed. Complete resection was achieved in 85–100%, with vital tumors in 61–100%. Where reported, the 90-day mortality rate was 0–11.4%. The reported survival metrics varied but included a median survival time (MST) 9–46 months and a 5-year OS rate of 20–75%. Recently, Romero et al. reported about 27 patients who underwent surgical resection after CRT. Complications were observed in 5 (18.5%) patients. The 3-and 5-year OS rates were 57.8% and 53.3%, respectively [19]. Furthermore, Kobayashi et al. reported 23 cases that underwent salvage surgery after CRT in a single center, with no perioperative death, a 5-year recurrence-free survival rate of 17.3%, and a 5-year OS rate of 41.9% [20]. Based on this evidence regarding salvage surgery after CRT, perioperative mortality appears to be acceptable, and long-term survival is possible in selected patients.

### 3.2. Salvage Surgery after Definitive Radiotherapy

Radiation monotherapy is indicated for patients with stage III NSCLC who are unsuitable for CRT. Stereotactic body radiotherapy (SBRT) is a good indication, especially for local lesions such as stage I NSCLC. Salvage surgery after definitive radiotherapy is more localized than definitive CRT. Since there is minimal effect on normal tissues, especially in SBRT and heavy-ion radiotherapy, the incidence of complications is expected to be low. In 2018, Dickhoff et al. performed a systematic review of salvage surgery after local recurrence of NSCLC after SBRT (7 case series with a total of 47 patients) [21]. The 5-year local recurrence rate after SBRT was approximately 10% and surgery was performed as salvage surgery in selected patients. The morbidity rate was 29–50%, and the 90-day mortality rate was 0–11%. MST ranged between 13.6 and 82.7 months. In addition, 12 patients who underwent salvage surgery after heavy-ion radiotherapy were reported by Mizobuchi et al. in 2015 [22]. There were no serious complications in any of the cases, and the 3-year survival rate after surgery was 82%. Although there is only limited evidence regarding salvage surgery after radiotherapy for locally relapsed NSCLC, this treatment can be considered feasible and can provide acceptable morbidity and mortality rates for selected patients.

### 3.3. Salvage Surgery after Combination Therapy with CRT and Immunotherapy

In the phase III PACIFIC study, eligible patients received durvalumab after CRT, and this combination therapy significantly prolonged progression-free survival (PFS) compared with that in the placebo group (16.8 months versus 5.6 months) [23].

Regarding the addition of surgery to this combination therapy, the significance of surgical intervention after CRT followed by ICI remains unclear. Recently, a clinical trial (JCOG1807C) was initiated to clarify the safety and efficacy of multimodality treatment of pre- and postoperative durvalumab therapy after preoperative CRT for resectable superior sulcus tumor (SST) and durvalumab maintenance therapy after CRT for unresectable SST. In this study, eligible patients were assigned to two groups: concurrent CRT (cisplatin+S-1+radiotherapy 66 Gy) + two courses of durvalumab followed by surgery and adjuvant durvalumab for resectable SST and CRT followed by maintenance durvalumab for unresectable SST. The primary end-point is 3-year OS. We await the results of this trial.

## 4. Surgical Intervention for Stage IV Lung Cancer

### 4.1. Surgical Intervention for Patients with Oligometastatic NSCLC

Considering the indications for surgical intervention in oligometastatic NSCLC cases, brain and adrenal metastases as oligometastatic organs have been reported to have a relatively good prognosis with 5-year OS rates of 20% and 20–30% [24]. The ACCP guidelines state that in cases of single brain metastasis and adrenal metastasis, cN0–1 is indicated for local treatment of metastatic lesions and resection of the primary lesion [3]. In addition, the NCCN guidelines recommend local treatment for metastatic lesions and multidisciplinary treatments, including systemic treatment, for primary lesions in cases of single brain metastases [1]. There are two strategies including surgery for treating oligometastasis: (1) resection of the primary tumor in advance and then control of distant tumors using surgery/RT and micrometastasis with drug therapy, and (2) addition of local treatment (surgery/RT) for patients with residual tumors that responded to drug therapy and became localized; i.e., a salvage approach.

The efficacy of upfront resection of a primary lesion of oligometastatic NSCLC was reported by Wang et al. in 2018. They conducted a retrospective study of patients with oligometastatic NSCLC, and 172 patients were divided into two groups: group A underwent primary surgical treatment and adjuvant chemotherapy, while group B was treated with systematic chemotherapy and local RT. The MSTs in groups A and B were 48 months and 18 months, respectively, and the 5-year survival rates were 21.1% and 7.6%, respectively (*p* < 0.05). They concluded that the local surgical treatment of primary lesions of NSCLC significantly increased OS and the 5-year survival rates of patients with oligometastatic NSCLC [25].

Gomez et al. reported the efficacy of a salvage approach (the addition of local treatment after definitive drug therapy) for oligometastatic NSCLC in their phase II RCT. First-line therapy was four or more cycles of platinum doublet therapy or 3 or more months of EGFR or anaplastic lymphoma kinase (ALK) inhibitors. The locations of oligometastases were as follows: 13 brain, 10 bone, 8 adrenal gland, 7 pleura, 6 lung, 4 cervical lymph node, 2 liver, 2 spleen, 1 retroperitoneal lymph node, 1 paraspinal mass, and 1 kidney. After receiving first-line therapy, patients were randomly assigned to either a local consolidative therapy group (RT and/or surgery) or a maintenance treatment group. This study was terminated early after randomization of 49 patients. Among patients administered local consolidative therapies, 96% underwent some form of RT. The median PFS in the local consolidative therapy group was 11.9 months versus 3.9 months in the maintenance treatment group (hazard ratio 0.35, *p* = 0.0054). Furthermore, no grade 4 or 5 toxicities were reported. They suggested that the addition of local therapy after first-line therapy might improve PFS of patients with oligometastatic NSCLC [26].

With regard to the optimal modality of local treatment for oligometastatic NSCLC, to date, no RCTs have compared SBRT and surgery. Otake and Goto reviewed salvage SBRT for oligometastatic NSCLC and concluded that SBRT appeared to provide a high level of local control with minimal associated toxicity [27]. Although surgery is a powerful local treatment, pre-treatment and/or post-treatment as a combined-modality approach is often required for oligometastatic NSCLC. It is necessary to carefully select surgery or RT as local treatment, considering the patient’s ability to tolerate total therapies.

### 4.2. Salvage Surgery after Definitive Systemic Therapy

#### 4.2.1. Salvage Surgery after Treatment with TKIs

For patients with stage IV NSCLC, chemotherapy resulted in only approximately a 7% improvement in 1-year survival compared with survival with best supportive care [28]. Compared with chemotherapy, EGFR-TKI administration results in a high response rate in patients with *EGFR* mutation-positive lung cancer; in particular, osimertinib has a response rate of 80% [29]. Among patients in whom TKI has a dramatic effect, salvage surgery for local residual/recurrent lesions is a possible treatment strategy.

Hishida et al. reported nine patients with stage IV NSCLC who underwent tumor resection after gefitinib administration. Surgery was performed for local tumor persistence, recurrence, or re-growth after treatment with gefitinib (duration of administration, 2–36 months), and the median OS after resection was 32 months. The median recurrence-free period was reported to be 6 months [30]. In another study, Hishida et al. reported the long-term outcome of 4 patients who underwent pulmonary resection for residual/regrown primary lesion of NSCLC treated with gefitinib. Recurrence was observed in three of four cases; however, all of them survived for 5 years or more after surgery. The remaining case continued to receive TKI administration for 4 years after surgery without cancer relapse [31]. Based on these reports, although no large-scale data are available, it is quite possible that salvage surgery after EGFR-TKI can be expected to have local control effects. A similar significance of salvage surgery has been reported in a case report on the use of ALK inhibitors for NSCLC with *ALK* gene translocations [32].

Recently, the effectiveness of osimertinib as a postoperative adjuvant therapy for resectable *EGFR* mutation-positive NSCLC was reported as the result of a phase III trial (ADAURA Clinical Trials, NCT02511106) [33]. In that trial, disease-free survival was significantly longer among patients who were administered osimertinib than among those who received placebo (90% versus 44%). It was unclear whether TKI should be continued after salvage surgery following treatment using TKI. However, considering the result of the ADAURA trials, it is possible that the prognosis will be improved if TKI is continued even after complete resection by salvage surgery.

#### 4.2.2. Salvage Surgery after Treatment with ICIs

Recently, the effectiveness of the PD-1 inhibitor pembrolizumab in patients with stage IV lung NSCLC was reported as the result of a phase III trial (KEYNOTE-407). The estimated 6-month OS rate was 80.2% with pembrolizumab administration. Moreover, the median PFS with pembrolizumab of 10.3 months (not reached) was superior to that with platinum doublet chemotherapy (6 months) [34].

Furthermore, ICI combined with chemotherapy for stage IV NSCLC was reported to significantly improve OS and PFS compared with chemotherapy alone as reported in phase III trials (KEYNOTE-189 [35,36] and KEYNOTE-407 [37]). The median OS and PFS were 15.9–22 months and 6.4–8.8 months, respectively. These survival benefits were consistent regardless of the level of programmed death-ligand 1 expression. Based on this evidence, ICI or ICI combined with chemotherapy is recommended as standard therapy for stage IV NSCLC without a driver mutation.

Although there have been few reports concerning salvage surgery after ICI, Bott retrospectively examined 19 patients who underwent lung resection after ICI for metastatic or unresectable cancer, including lung cancer (47%) and metastatic melanoma (37%). Of patients who underwent resection, R0 resection was achieved in 95% and 68% of patients with viable tumor remaining. Complications occurred in 32% of patients. The 2-year OS was 77% [38]. In NSCLC, the frequency of salvage surgery has been increasing in recent years, and with the spread of ICI, salvage surgery after ICI may increase in the future. Salvage surgery after ICI is possibly a feasible and effective treatment; however, to date, it has only been described in one case report [39]. It is necessary to accumulate further evidence regarding salvage surgery after ICI.

Although there is no evidence that salvage surgery after definitive therapy confers a survival benefit compared with other non-surgical radical therapy, here, we show selected studies regarding current standard first-line therapy for stage III/IV NSCLC and limited reports on salvage surgery (Table 1).

## 5. Future Perspectives

Several clinical questions, such as whether surgery or RT is better as a local salvage treatment after definitive systemic therapy or as the first-line local treatment for oligometastatic NSCLC, remain unanswered. The significance of surgical intervention after CRT followed by ICI treatment is also unknown. The use of adjuvant therapy was not described in this review; furthermore, the chronological time of TKI/ICI addition to surgery remains yet to be demonstrated (adjuvant or neoadjuvant). In addition, it is possible that the definition of “unresectable” may change, and R1/R2 resection or even volume reduction may turn out significant in the future. Prospective and comparative trials need to be performed to clarify these issues in the future.

## 6. Conclusions

This review covered the role of surgical intervention for unresectable NSCLC. Although many problems still need to be solved, while systemic treatments with high response rates such as TKI and/or ICI are being developed further, the importance of surgical treatment is expected to expand.

## Figures and Tables

**Figure 1 jcm-09-03881-f001:**
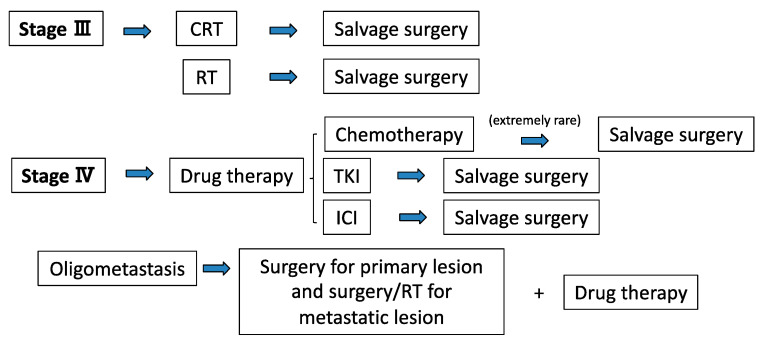
Current treatment strategies with surgery for unresectable stage III/IV NSCLC. CRT: chemoradiotherapy, ICI: immune-checkpoint inhibitor, NSCLC: non-small cell lung cancer, RT: radiotherapy, TKI: tyrosine kinase inhibitor.

**Table 1 jcm-09-03881-t001:** Selected studies on salvage surgery and first-line therapy for stage III/IV NSCLC.

Stage	Modality	OS (%), MST (m)	mPFS (m)	RR (%)	Ref.
Stage III	CRT only	5-y OS: 5–25, 26	-	-	[6,15]
	CRT → ICI	3-y OS: 66.3, 38.4	16.8	28.4	[22]
	CRT → SS	5-y OS: 20–75, 9–46	-	-	[18]
	RT → SS	N.A., 13.6–82.7	-	-	[21]
Stage IV	EGFR-TKI only(osimertinib)	1.5-y OS: 83, N.A.	8.9	80	[28]
	ICI only(pembrolizumab)	0.5-y OS: 80.2, 14–19.2	10.3	44.8	[33]
	ICI + CT	2-y OS: 45.7, 15.9–22	6.4–9	48	[34,35,36]
	EGFR-TKI → SS	3-y OS: 50, 32	-	-	[29]
	ICI → SS	2-y OS: 77, N.A.	-	-	[37]
(Oligometastasis)	CT + RT (all lesions)	5-y OS: 7.6, 18	-	-	[24]
	Surgery (all lesions) → CT	5-y OS: 21.1, 48	-	-	

CRT: chemoradiotherapy, CT: chemotherapy, EGFR-TKI: epidermal growth factor receptor-tyrosine kinases inhibitor, ICI: immune-checkpoint inhibitor, mPFS: median progression-free survival, MST: median survival time, N.A.: not available, NSCLC: non-small cell lung cancer, OS: overall survival, Ref.: reference, RR: response rate, RT: radiotherapy, SS: salvage surgery, y: year.

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
