# Peer review of "Role of Surgical Intervention in Unresectable Non-Small Cell Lung Cancer"

_jcm, 2020, doi:10.3390/jcm9123881_

Round 1

Reviewer 1 Report

The authors must be complemented with addressing a topic of interest at the moment, since knowledge about treatment of stage III and IV in evolving rapidly and the role of surgery is changing.

Focus of the manuscript is on technically difficult resections (unresectable in a narrow sense), without the discussion about whether or not resection is indicated.

The paper is well written without hardly any grammatical errors so, in my opinion, it needs only little editing.

Several reports on this topic have been published the past few years, so the message is not new, unfortunately. Moreover, the information in this review is limited and incomplete. However, the highlights of available knowledge are summarized in this manuscript,

Some references are outdated (Ref 7, 8, 9) and might be replaced by recent references with for example ‘modern’ radiotherapy schedules.

References about salvage surgery are omitted (lines 75-76 and 78-79), nor up to date and complete. It might be considered to also include:

  • Dickhoff C, Dahele M, Paul MA, et al. Salvage surgery for locoregional recurrence or persistent tumor after high dose chemoradiotherapy for locally advanced non-small cell lung cancer. Lung Cancer 2016;94:108-113.
  • Sonobe M, Yutaka Y, Nakajima D, et al. Salvage Surgery After Chemotherapy or Chemoradiotherapy for Initially Unresectable Lung Carcinoma. Ann Thorac Surg 2019;108:1664-1670.

I agree with the assumption mentioned in lines 103-104, however, references are needed to confirm this statement.

References on the topic oligometastasis might be updated with the in 2019 published paper on this topic from the EORTC Lung Cancer Group (Eur J Cancer, Dingemans et al.).

In Table 1, references should be added.

In the conclusion, (lines 255-256) some statements are done, not earlier mentioned in the manuscript, with possible references lacking unfortunately.

Author Response

Comment 1

Some references are outdated (Ref 7, 8, 9) and might be replaced by recent references with for example ‘modern’ radiotherapy schedules.

Response:  According to the reviewer’s suggestion, References 7 and 9 were updated and new reference was added to Reference 8.

Comment 2

References about salvage surgery are omitted (lines 75-76 and 78-79), nor up to date and complete. It might be considered to also include:

  • Dickhoff C, Dahele M, Paul MA, et al. Salvage surgery for locoregional recurrence or persistent tumor after high dose chemoradiotherapy for locally advanced non-small cell lung cancer. Lung Cancer 2016;94:108-113.
  • Sonobe M, Yutaka Y, Nakajima D, et al. Salvage Surgery After Chemotherapy or Chemoradiotherapy for Initially Unresectable Lung Carcinoma. Ann Thorac Surg 2019;108:1664-1670.

Response: As the reviewer suggested, these references were added to the manuscript (line 76).

Comment 3

I agree with the assumption mentioned in lines 103-104, however, references are needed to confirm this statement.

Response: As the reviewer suggested, two references were added to this sentence.

Comment 4

References on the topic oligometastasis might be updated with the in 2019 published paper on this topic from the EORTC Lung Cancer Group (Eur J Cancer, Dingemans et al.).

Response: As the reviewer suggested, this reference was added to the manuscript (line 87).

Comment 5

In Table 1, references should be added.

Response: The references are shown at the last column in Table 1. To clearly show these references, they are now put in parentheses.

Comment 6

In the conclusion, (lines 255-256) some statements are done, not earlier mentioned in the manuscript, with possible references lacking unfortunately.

Response: According to the reviewer’s suggestion, this statement was moved to the section “Future Perspectives”.

Thank you for your thoughtful comments.

Reviewer 2 Report

This manuscript is a review entitled “Role of surgical intervention in unresectable non-small cell lung cancer”.

The authors review the recent literature of the therapeutic approaches for the stage III/IV non-small cell lung cancer (NSCLC) focusing on the role of surgical intervention. As the authors clarify in the introduction, the review focuses on the patients with “unresectable” NSCLC, i.e., patients with NSCLC for whom the complete resection is technically difficult.

The topic is interesting, and the manuscript is quite well written and compact. The authors refer to a sufficient amount of publications.

In some parts, the font changes in a somewhat arbitrary manner (italics) that should be corrected to ease the readability of the review. Otherwise, I do not have any comments to improve the manuscript.

Author Response

Comment 1

In some parts, the font changes in a somewhat arbitrary manner (italics) that should be corrected to ease the readability of the review.

Response: Thank you for your kind comments on my paper. The phrases or sentences with wrong italic font were corrected, as per the reviewer’s suggestion.